# A Comprehensive Evaluation of Carbon Emission Reduction Capability in the Yangtze River Economic Belt

**DOI:** 10.3390/ijerph17020545

**Published:** 2020-01-15

**Authors:** Decai Tang, Yan Zhang, Brandon J Bethel

**Affiliations:** 1China Institute of Manufacturing Development, Nanjing University of Information Science & Technology, Nanjing 210044, China; tangdecai@nuist.edu.cn; 2School of Management Science and Engineering, Nanjing University of Information Science & Technology, Nanjing 210044, China; 3School of Marine Sciences, Nanjing University of Information Science & Technology, Nanjing 210044, China; bjbethel09@gmail.com

**Keywords:** Yangtze River Economic Belt, carbon emission reduction capacity, obstacle factor, emission reduction path

## Abstract

The Yangtze River Economic Belt (YREB) is an essential part of China’s goal of reducing its national carbon emissions. Focusing on economic and social development, the development of science and technology, carbon sinks, energy consumption, and carbon emissions, this paper uses “the Technique for Order of Preference by Similarity to Ideal Solution mode” (TOPSIS) and “an obstacle factor diagnosis method” to measure the reduction capacity of each province and municipality of the YREB. Key obstacles to achieving the goal of carbon emission reduction are also identified. The main finding is that the emission reduction capacities of Shanghai, Jiangsu and Zhejiang in China’s east is far greater than that of all other provinces and municipalities, the main obstacle of Shanghai, Jiangsu, and Zhejiang are carbon sinks, energy consumption and carbon emission, and other provinces and municipalities are social and economic development. Taking into consideration those evaluation results and obstacles, paths for carbon emission reduction are delineated through a four-quadrant matrix method with intent to provide suitable references for the development of a low-carbon economy in the YREB.

## 1. Introduction

Low-carbon economy has become the basic direction of the world economic development, energy conservation and emission reduction are the main means to respond to climate change. In recent years, the explosive growth of China’s carbon emissions has been curbed due to the introduction and promotion of relevant policies (Energy conservation and emission reduction policy measures include signing climate change declarations with many countries, promoting China’s emission reduction targets to the world, and launching the national carbon market). The newest research has indicated that carbon intensity, which is defined as the emission rate of a given pollutant relative to the intensity of a specific activity or an industrial production process, has been in 2018 reduced approximately 45.8% lower than as compared to 2005, meeting and marginally exceeding the original 2020 target of 40–45%.

The Yangtze River Economic Belt (YREB) contains 11 provinces and municipalities which include Shanghai, Jiangsu, Zhejiang, Anhui, Jiangxi, Hubei, Hunan, Chongqing, Sichuan, Yunnan, and Guizhou. As can be seen from Figure 1, the carbon emission of the YREB (In the article “An Analysis of Disparities and Driving Factors of Carbon Emissions in the Yangtze River Economic Belt”, the author adopted the inventory method given by the International Panel on Climate Change (IPCC), and selected raw coal, coke, crude oil, fuel oil, gasoline, kerosene, diesel, and natural gas to calculate the carbon emissions of the YREB) generally showed a continuous increasing trend from 2005 to 2016, with the carbon emission increasing from approximately 700,000 tons to just over a million tons. Although the growth rate of carbon emissions has slowed down since 2011, the base of carbon emissions is large. Following the designation of the YREB as one of the country’s three regional development strategies (The three major strategies for China’s regional development include: “One Belt and One Road”, coordinated development of Beijing-Tianjin-Hebei, and the YREB) the problem of how to achieve low-carbon development has received increasing scrutiny from the national government and scholars. Therefore, an empirical study on carbon emission reduction capacity (CERC) of the YREB can not only enrich the pre-existing carbon emission reduction performance evaluation index system, but also provide a theoretical foundation and a reference for decision-makers in the development of a low-carbon economy in the YREB. This paper is structured as follows: Section 2 contains the literature review, Section 3 and Section 4 contain the research methodology and empirical research results, respectively. Section 5 presents a conclusion and discussion.

## 2. Literature Review

Recently, energy conservation and emission reduction are two frontier problems in energy economics research. Literature on the subject broadly falls into two categories: (1) the regional comprehensive assessment of the factors affecting regional energy conservation (EC) and emission reduction (ER) and (2) EC and ER development level. Influencing factors research can be further subdivided into single-element and total factor frameworks.

Early research centered around the single-element framework, i.e., using the total amount or intensity index such as carbon emissions and carbon intensity to measure the energy utilization level of the region and analyze its influencing factors. Fisher [1], using data acquired from Chinese industrial enterprises over the 1997 to 1999 period, concluded that the main reason for the decline in China’s absolute energy use and intensity was due to increases in energy prices, the strengthening of research and development and the conclusion of enterprise ownership reform. Fan et al. [2], arrived at a similar conclusion of declining energy intensity by using data which ranged from 1980–2003. Those researchers found a decrease in CO_2_ emission intensity, which was attributed to the decline in energy intensity. Amongst the factors which can affect carbon intensity including energy intensity, economic activity and structure, and CO_2_ emission coefficient, Zhang et al. [3] indicated that the most important of these for China was the energy intensity. 

In contrast to the single-element framework, total factor framework introduces more factors such as resources, the environment, and living standard to comprehensively examine the regional development of emission reduction. For example, Zhou et al. [4] used an environmental Data Envelopment Analysis-based Malmquist Index to measure the performance of the world’s top 18 emitters’ emission reduction and the dynamic influencing factors. Liu et al. [5] selected 13 indicators based on the carbon source/sink analysis framework to empirically study the low-carbon development level of 36 low-carbon pilot cities and found that Xiamen and Jiyuan have the highest and lowest levels of carbon development respectively. Wu [6] constructed a theoretical energy efficiency stochastic frontier model based on the Shandong Province industrial total factor energy efficiency to estimate 37 industrial sectors’ panel data over the 2006–2013 period and found that the scale of enterprises, the degree of openness, the level of foreign investment, technological innovation, and energy efficiency are significantly positively correlated, while the proportion of state-owned and state-controlled enterprises is significantly negatively correlated with energy efficiency. Deng et al. [7] used the differential internal difference method to study the impact of the Northeast Revitalization Strategy and the Western Development Strategy on the total factor energy efficiency of China under the environmental constraints over the 1987–2012 period, identifying that Northeast Revitalization project is committed to transforming the original high-energy and extensive growth mode and improving the energy efficiency of the northeast, but Western Development project is mainly based on the development of resource-based industries, and has not improved the energy efficiency of the western region. 

A series of studies on the evaluation of EC and ER development mainly include the design of evaluation index systems and their accompanying evaluation models. For the design of an evaluation index system, there are few examples available in the literature. Although a standardized evaluation index system has yet to be devised, the Chinese Academy of Social Sciences in 2010 released China’s first relatively complete low-carbon economic evaluation standard system (The standard system for assessing low-carbon cities is divided into four categories: low-carbon productivity, low-carbon consumption, low-carbon resources, and low-carbon policies. If a city’s low-carbon productivity indicator exceeds 20% of the national average, it can be considered “low-carbon”). Su and Liang [8] proposed an evaluation index system for urban low-carbon development from the perspective of economic development and social progress, energy structure and usage efficiency, living consumption, and development surroundings, and selected 12 typical Chinese cities as cases studies, identifying that 12 cities can be classified into three groups in terms of their low-carbon development: (1) The first group is a large city with a low level of low-carbon development, and the relatively slow development speed and development environment are the limiting factors. (2) The second category is coastal cities with medium or low-carbon development levels, medium growth rates, and living consumption constraints. (3) The third category is inland cities, which are characterized by relatively low levels of low-carbon development, relatively rapid development, and constraints on economic development and social progress. Similarly, Qu and Cao [9] first considered the indicators of people’s livelihood and constructed a multi-level indicator system with low-carbon output, emissions, consumption, in addition to people’s living standards as the first-level indicators. These first-level indicators were applied to the evaluation of the development level of low-carbon economy in Shaanxi Province. There, research found that the low-carbon economy in Shaanxi Province has a low level of development compared with the national average. For the construction of an evaluation model, the research ideas are relatively unified. The first type of method is to determine the weight of each index through an analytic hierarchy process. It has the advantages of simple operation and strong applicability, but the reliability of the model needs to be improved. The second type of method is the improvement of the analytic hierarchy process, embodying the idea of multidisciplinary integration. Jia et al. [10] introduced the Technique for Order of Preference by Similarity to Ideal Solution mode (TOPSIS) method into the low-carbon economic evaluation, considering the ambiguity and nonlinear problems in the evaluation, and using fuzzy analytic techniques to determine the index weights. Guo [11] used the index increment and temporal entropy-weighting to improve the traditional static TOPSIS method to evaluate regional low-carbon economy competitiveness and analysis its space difference. That researcher found that the dynamic TOPSIS method can not only simultaneously compare evaluation results of different objects, but also simultaneously indicate the evolution of the same evaluated object. Wang [12] established a collaborative model for system innovation and green transformation analysis, and dynamically assessed the ability of regional green transformation. He found that the overall coordination between system innovation and green transformation in the Jiaodong Peninsula was good from 2010 to 2013, but system innovation capabilities cannot be fully transformed into green transformation capabilities.

Although the academic community has conducted qualitative and quantitative research on the performance evaluation of energy saving and emission reduction, there are still some problems. Firstly, the design of the evaluation index system of low-carbon cities needs further improvement. Secondly, there is a dearth of literature on the comprehensive evaluation of carbon emission reduction in the YREB. This paper takes 11 provinces and municipalities in the YREB as the research object, tries to construct a reasonable carbon emission reduction comprehensive evaluation index system, examines the carbon emission reduction capacity of each province/municipality, introduces a four-quadrant analysis method to analyze the emission reduction path, and puts forward the corresponding countermeasures and suggestions.

## 3. Research Methodology

### 3.1. Entropy Weight TOPSIS

The TOPSIS method was proposed by C. L. Hwang and K. Yoon in 1981. It sorts according to the closeness of a limited number of evaluation objects to the idealized target. It is a sorting method that approximates the ideal solution. The TOPSIS method of entropy is the improvement on the traditional TOPSIS evaluation method. Its construction is as follows: firstly, the entropy weight method is used to determine the weight of the evaluation index, and then the technique of approximating the ideal solution is used to determine the ranking of the evaluation object. The entropy weight method is based on the information provided by each evaluation index. This allows for not only an objective reflectance of the relative importance of a given indicator within the indicator system, but also allows for the precise measurement in the change of index weight over time. The core idea of the TOPSIS method of entropy is to define the direct distance between the optimal solution and the worst solution of a given decision problem. Then, the relative progress of each solution and its corresponding ideal solution is calculated. The pros and cons of each solution are consequently ranked. Using the TOPSIS method of entropy to determine weight is an important part of this current study and by using the method of information entropy, the effective elimination of the influence subjective of factors becomes possible. 

### 3.2. Obstacle Factor Diagnostic Model

The purpose of the comprehensive evaluation of CERC is to identify gaps and improve the direction and to diagnose the barriers that affect carbon emission reduction so that the appropriate countermeasures can be developed. Therefore, this paper introduces an obstacle model to conduct an extended study on carbon emission reduction. The obstacle degree calculation uses three factors in its diagnosis: factor contribution, index and obstacle degrees. Firstly, the factor contribution $\omega_{j}$ is the contribution of the single indicator to the total carbon emission reduction target. Secondly, the indicator deviation degree $O_{j}$ is the difference between the actual value of the individual indicator and the optimal target value, which is set as the standardized value of the individual indicator and is 100%. Thirdly, the obstacle degree $I_{j}$ is the degree of influence of subsystems or individual indicators on CERC. The formula is as follows:(1)$$O_{j}=1-x_{ij}^{\prime }$$
(2)$$I_{j}=\frac{O_{ij}\times \omega_{j}}{\sum_{i=1}^{n}O_{ij}\times \omega_{j}}$$

## 4. Empirical Research—Comprehensive Evaluation of CERC in the YREB

### 4.1. Construction of the Index System

In empirical research, the construction of a comprehensive evaluation index system for systems science is key, and the related research are shown in Table 1. Yao and Ni [13] divided five subsystems when evaluating regional carbon emission reduction capabilities: industry and energy consumption structure, opening up, technology and carbon sinks, energy consumption and carbon emissions, economic development. In assessing emission reduction potential, Wu [14] divided into three subsystems: emission reduction responsibility, emission reduction capability, and difficulties of emission reduction. When evaluating the low-carbon economic development, Su and Chen [8] divided four subsystems, economic development and social progress, energy structure and usage efficiency, living consumption, development surroundings, and Guo [11] also divided four subsystems, economic developmental, social development, environmental assessment, scientific. Based on the composition of the above-mentioned literature subsystems, and taking into account the actual development of carbon emission reduction in the YREB and the availability of data, this article screens out five subsystems of economic development, social development, technological development, energy consumption, and environmental development. Finally, shown in Table 2, a CERC evaluation index system containing five first-level indicators and 26 second-level indicators was constructed to reflect the development status of CERC of 11 provinces and municipalities in the YREB. Basic data is derived from the 2006, 2010, and 2017 China Statistical Yearbooks. Price-related data are calculated at constant prices in the year 2000.

### 4.2. CERC Evaluation Analysis Results

#### 4.2.1. Comprehensive Evaluation of CERC

The comprehensive evaluation results of CERC of the 11 provinces and municipalities in the YREB for 2005, 2010, and 2016 can be computed by the TOPSIS method of entropy (Figure 2). In addition, the standard differential level method is used to divide each progress into three levels: low, middle, and high. Arc GIS 10.2 (Environmental Systems Research Institute, Inc., Redlands, CA, USA) is used for spatial statistical analysis to describe the distribution of carbon emission reduction capacities of each province /municipality (Figure 3). 

As is shown in Figure 2, analysis from time scale, the comprehensive capacity of carbon emission reduction showed a different range of increase or decrease trends in the 11 provinces and municipalities in the years of 2005, 2010, and 2016. Shanghai and Zhejiang showed a slight downward trend, which may be related to the high ability of carbon emission reduction in the initial stage, the limited room for promotion, and the rapid growth of carbon emissions caused by the rapid economic development. Other provinces and municipalities have shown different growth rates. Among them, Chongqing’s carbon emission reduction comprehensive capacity has the largest increase, which increased by 119.5% in 11 years. The reason may be that the initial CERC is too low, so the promotion space is large, and the starting time of the rapid development of Chongqing is relatively late, so the carbon emissions of energy consumption are relatively low. Sichuan’s carbon emission reduction comprehensive capacity has the smallest increase, which has increased by 5% in 11 years, and the reason may be that the slowdown carbon sink and slow growth of governance input. In the remaining 11 provinces and municipalities, the comprehensive capacity improvement of carbon emission reduction was between 10% and 20%, and the growth rate was not large. 

As can be seen in Figure 3, there is a disparity of CERC between the provinces and municipalities of the YREB. Specifically, in 2005, Jiangsu, Shanghai, and Zhejiang could be classified as high-level areas whereas the remaining provinces and municipalities were all low-level areas. In 2010, by contrast, the only provinces that were classified as high-level areas were Jiangsu and Shanghai, with Zhejiang listed as a middle-level area; the remaining provinces and municipalities were classified as low-level areas. In 2016, Jiangsu and Shanghai were high-level areas, Zhejiang and Chongqing were middle-level areas, and the remaining provinces and municipalities were low-level areas; the average situation for three years is consistent with that for 2010. Generally, the carbon emission reduction ability of the YREB from 2005 to 2016 is strong in the east and weak in the central and western regions, and the gap between provinces and municipalities shows a narrowing trend from the point of view of standard deviation. The reasons may be as follows: firstly, there exists significant disparities in China’s regional development and natural resource endowments, resulting in a gap between total carbon emissions and carrying capacity; secondly, during the period of the “Eleventh Five-Year Plan” in effect from 2006 to 2010, China established a low-carbon development policy that aimed at promoting the development of new energy sources and renewable energy, saving energy and enhancing the capacity of carbon sinks.

Furthermore, we can observe that the absolute difference between the provinces or municipalities with the highest and lowest carbon emission reduction capacities decreased from 0.532 in 2005 to 0.449 in 2016. Perhaps the reduction in absolute difference can be strongly linked with strengthened exchanges and cooperation between provinces and municipalities.

#### 4.2.2. Evaluation of Carbon Emission Reduction Capability of Subsystem

To further analyze the possible reasons for the observed trend of CERC in various provinces and municipalities in the YREB, this paper calculates the carbon emission reduction ability scores of five subsystems and analyzes their characteristics and evolution trends. Results are shown in Figure 4. In the years of 2005, 2010, and 2016, the scores of the subsystems of the 11 provinces and municipalities in the YREB showed different growth trends:(1)In the economic development subsystem, Shanghai, Zhejiang, Hubei, showed a slight decline, Jiangsu, Anhui, Jiangxi, Hunan, Chongqing, Sichuan, Guizhou, and Yunnan are on the rise. From the perspective of spatial distribution, the economic development capacity of the YREB is strong in the east and weak in the central and western regions. The absolute difference of economic development capacity between the first and last provinces was marginally reduced from 0.671 to 0.621.(2)In the science and technology subsystem, Shanghai, Jiangxi, Hunan, and Chongqing showed a slight decline. Jiangsu, Zhejiang, Anhui, Hubei, Sichuan, Guizhou, and Yunnan showed an upward trend. From the perspective of spatial distribution, the technological development capability of the YREB is strong in the east and weak in the central and western regions. The absolute difference of science and technology development capacity between the first and last provinces was reduced from 0.692 to 0.642.(3)In the carbon sink subsystem, Jiangsu and Zhejiang showed a slight decline. Shanghai, Anhui, Jiangxi, Hunan, Chongqing, Sichuan, Guizhou, and Yunnan showed an upward trend. From the perspective of spatial distribution, the carbon sink development capacity of the YREB is strong in the central and western regions, followed by the eastern part. The absolute difference of the carbon sink development capacity between the first and last provinces was reduced from 0.623 to 0.477.(4)In the energy consumption and carbon emission subsystem, Shanghai, Hunan, and Yunnan showed a slight decline. Jiangsu, Zhejiang, Anhui, Jiangxi, Hubei, Chongqing, Sichuan, and Guizhou showed an upward trend. From the perspective of spatial distribution, the energy consumption and carbon emission capacity of the YREB is strong in the east and west, and followed by the central part. The absolute difference of energy consumption and carbon emission capacity rose from 0.417 to 0.502.(5)In the social development subsystem, Shanghai and Zhejiang showed a slight decline. Jiangsu, Anhui, Jiangxi, Hubei, Hunan, Chongqing, Sichuan, Guizhou, and Yunnan showed an upward trend. From the perspective of spatial distribution, the energy consumption and carbon emission capacity of the YREB is strong in the east and followed by the central and western regions. The absolute difference of energy consumption and carbon emission capacity between the first and last provinces was reduced from 0.692 to 0.413.

### 4.3. Obstacle Factors Diagnosis 

Combined with the above comprehensive evaluation results, the Equations (1) and (2) are used to determine the obstacles affecting the various subsystems and indicators of the CERC of the YREB (Figure 5 and Figure 6).

#### 4.3.1. Subsystem Barrier Factor

For brevity, only the first influencing factor of obstacle degree is shown in Figure 5. We can observe there are some differences in the main barrier factors of CERC between provinces and municipalities at different points in the YREB. In the years of 2005, 2010, and 2016, the main obstacle to Shanghai, which belongs to the high-level areas, was the carbon sink subsystem, and its obstacle degree increased from 52 to 79.36. Therefore, Shanghai should strengthen the protection of the ecological environment and improve the carbon sink capacity. The main obstacles in Jiangsu Province, which belongs to the high-level areas, were the energy consumption and carbon emission or social development subsystems, and its obstacle degrees have dropped from 47.69 to 41.94. Therefore, Jiangsu Province should maintain its advantages and focus on improving energy utilization efficiency and social development level. The main obstacles in Zhejiang Province, which belongs to the middle-level areas, were energy consumption and the carbon emission or economic development subsystems. The obstacles increased from 42.25 to 42.54, so Zhejiang Province should pay attention to the development of low-carbon technologies and improve the quality of economic development. The main obstacles in the eight provinces and municipalities of Anhui, Jiangxi, Hubei, Hunan, Chongqing, Sichuan, Guizhou, and Yunnan, which belong to low-level areas, were the economic or social development subsystems. The degree of obstacles is between 64.27 and 98.68, showing an upward trend. The barriers between Guizhou and Yunnan were much larger than those of the other six provinces and municipalities. 

#### 4.3.2. Indicator Layer Barrier Factor

Similar to the above subsection, only the first influencing factors of obstacle degree is shown for brevity. According to Figure 6, the top two obstacle factors account for a substantial proportion of carbon sink and social development subsystems. From 2005 to 2016, according to the frequency of occurrence, the carbon absorption of crops (B14) and forest coverage (B11) have the greatest impact on the CERC in provinces/municipalities classified as middle-levels and high-levels. Consequently, we may infer that Jiangsu, Zhejiang, and Shanghai possess low-carbon sink capacities.

In low-level areas, coal consumption as a share of energy consumption (B19) has the greatest impact on CERC in the cases of Anhui, Jiangxi, and Guizhou, indicating that the energy consumption structure of these provinces still requires further optimization. Per capita park green area (B12) has the greatest impact on Hunan’s CERC, indicating that for Hunan province, it is similar to Jiangsu, Zhejiang and Shanghai, is also deficient from the perspective of its carbon sink capacity. Deficiency in product quality (B9) has the greatest impact on the CERC in Hubei and Chongqing, from which it may be inferred that the scientific and technological capability of these two areas should be improved. In the case of Sichuan, carbon footprint (B18) has the greatest impact on the CERC. By contrast to Sichuan, the city gas penetration rate (B26) has the greatest impact on CERC in Yunnan. Moreover, we can observe that the GDP growth rate (B2) and public transport vehicles per 10,000 people (B25) can also affect carbon emission reduction, signaling that both Sichuan and Yunnan should take steps to improve both the economic and social development capacities.

### 4.4. Carbon Emission Reduction Route Planning

By taking the CERC scores of each province/municipality in the YREB in 2005, 2010, and 2016 as the x-axis, and the weighted average degree of carbon emission reduction obstacles obtained by each subsystem as the y-axis, we can plot a four-quadrant matrix diagram to analyze the potential and path of carbon emission reduction (Figure 7). Each quadrant can be listed as follows:(1)Mature (Quadrant I): The provinces and municipalities in this quadrant have completed carbon emission reduction optimization reforms. Emission reductions have been significant and all obstacles in these areas are relatively small. Innovation should be encouraged, the existing advantages should be maintained, and the quality of development should be improved further.(2)Improving (Quadrant II): The provinces and municipalities located in this quadrant have certain advantages but are subject to the inertia of economic growth mode, there is still a large space for emission reduction. These provinces should make full use of its own capabilities to develop energy-saving and emission-reduction technologies to transform to a Mature model.(3)Developing (Quadrant III): The provinces and municipalities located in this quadrant are in the nascent stage of development. In these areas, economic strength is bought by sacrificing the ecological environment and then low-carbon transformation is carried out to actualize sustainable development. The specific path is a transformation from Quadrant III to Quadrant IV, then Quadrant II, and finally Quadrant I. However, this path is not only time consuming, but also likely to drive the further deterioration of the environment. To avoid this problem, the development mode transformation should be accelerated and achieve a truly green development based on the maintenance of the existing level of carbon emissions, and the specific path is a transformation from Quadrant III to Quadrant I.(4)Underdeveloped (Quadrant IV): The provinces and municipalities located in this quadrant have low economic productivity, high energy intensity, and large obstacles to carbon emission reduction. The specific path is a transformation from Quadrant IV to Quadrant II, then Quadrant I. However, in the current environment, the development requires both economic and ecological benefits. The development mode should be modified, the overcapacity (i.e., production capacity is greater than social demand) should be decomposed (i.e., excess production is discarded or modifying existing production capacities to produce other products), development should be accompanied by emission reductions.

## 5. Conclusions

This paper selected 11 provinces and municipalities of the YREB as the research object and used the TOPSIS method of entropy to dynamically present the carbon emission reduction development status of the YREB’s various provinces and municipalities in the years of 2005, 2010, and 2016. This research also analyzed the obstacle factors of carbon emission reduction to explore the emission reduction path. The specific research conclusions are as follows.
(a)From the perspective of comprehensive scores, there exists significant disparities in carbon emission reduction capacities between eastern, central and western regions. As a poignant example of this conclusion, we found that the emission reduction capacities of Shanghai, Jiangsu, and Zhejiang in China’s east is far greater than that of all other provinces and municipalities. From a dynamic perspective, the differences in CERC between each province/municipality are decreasing where middle-level areas presented a downwards trends. By contrast, low-level areas showed an upward trend. Here it can be highlighted that the interannual fluctuation range of CERC in Chongqing was significant. From the perspective of subsystem scores, the economic, technological, and social development capabilities are strong in the eastern region, followed by the central and western regions. The carbon sink capacities are strong in the central and western regions, followed by the eastern region. Energy consumption and carbon emissions are strong in the eastern and western regions, while small in the central region. Some scholars’ conclusions support our view. For example, Yao’s research also found that as economically developed regions, Jiangsu, Tianjin, and Zhejiang have economic and technological advantages, and have stronger potential for reducing emissions and reducing emissions [13]. Yu found that the regions with higher energy conservation and emission reduction efficiency are mainly concentrated in developed municipalities, Guangdong, Jiangsu, and other regions [15]. This is consistent with the distribution of carbon emission reduction capacity of the YREB.(b)From the perspective of subsystem barrier factors, the main obstacle factors of Shanghai, Jiangsu and Zhejiang are their respective carbon sinks, energy consumption and carbon emission. The main obstacle factors in other provinces and municipalities are social and economic development. Specifically, from barrier factor indicators, the main obstacles are varied, the most influential factors come from carbon sinks and social development subsystems, including crop carbon uptake, forest coverage, public transport vehicles per 10,000 people, and the urban gas penetration rate. There are some consistent conclusions in the study of carbon emission driving factors and carbon emission reduction efficiency. Economic development, industrial structure, energy consumption, and social development are all significant influencing factors [16,17]. In researching the low-carbon development model of typical cities in China, Su included that the limiting factors of Shanghai (representative of Beijing, Guangzhou, and Shanghai) were development surroundings and living consumption, those of Zhuhai (representative of Shenzhen, Qingdao, Hangzhou, Tianjin, and Zhuhai) were living consumptions and development surroundings, and those of Baoding (representative of Kunming, Chongqing, Suzhou, and Baoding) were economic development and social progress and development surroundings [8]. From the perspective of urban type, we find that the results of our study are consistent with that of Su.(c)Naturally, the various provinces and municipalities endowed by different emission reductions have different emission reduction potentials. The planning of emission reduction routes should consider both emission reduction capacity and obstacles. Shanghai, Jiangsu, and Zhejiang are mainly Mature and Improving, and the emission reduction path should be a transformation from Improving to Mature, which is a development method focusing on technological innovation to improve development quality. Other provinces and municipalities are mainly Underdeveloped, and emission reduction path should be a transformation from Underdeveloped to Improving, then Mature, which currently places their focus first on development, followed by emission reductions.

According to the research conclusions, the main revelations of this paper are as follows:(a)The YREB’s eastern region should bear a greater responsibility for reducing emission than the central or western regions as its capability to achieve this is not only greater than the other two regions, but given its historical carbon emissions, its responsibility for reductions is consequently right. In the less developed central and western regions, their carbon emission should be reduced based on the time required for economic development and industrial transformation.(b)Grasping the two key points of carbon source and carbon sink, the YREB should actively develop clean energy and vigorously develop low-carbon technologies to greatly increase carbon emission reductions [18,19,20]. Shanghai, Jiangsu, and Zhejiang should fully utilize their technical capabilities to replace the current dirty sources of energy (i.e., fossil fuels) with cleaner, renewable energy sources. For now, central and western regions should switch to cleaner burning coal, and gradually eliminate underdeveloped industries. Furthermore, increasing green investments, improving carbon sink capacities, and environmental self-purification capacities are all effective means to reduce carbon emissions [21,22,23].(c)To promote industrial upgrades and transfers, the eastern provinces and municipalities should promote independent innovation, increase production capacity, and promote low-carbon industrialization. Central provinces should eliminate underdeveloped production capacities and rely on technological innovation to reduce the dependence of economic development on resources. Western provinces and municipalities should increase investments in science and technology and undertake industrial transfer in developed regions. At the same time, exchanges and cooperation between technology industries should be strengthened between regions, so that regions with strong economic strength, technical talent levels will have a driving effect on weaker regions and promote a high-quality development.(d)Given that rapid urbanization leads to rapid growth of living carbon emissions, the promotion of low-carbon urban lifestyles should be at the forefront of future policies. The eastern provinces and municipalities have weak environmental carrying capacity and dense population, and so, low-carbon technologies and policy incentives should promote appropriate low-carbon changes in transportation and housing. Similarly, the social development capacity of the central and western provinces and municipalities is also weak. We should increase investment in education to enhance residents’ awareness of environmental protection, formulate low-carbon policies to promote low-carbon buildings, decrease household carbon emissions, and at the same time, build an optimized urban system and promote high-quality urbanization development [24,25].

## Figures and Tables

**Figure 1 ijerph-17-00545-f001:**
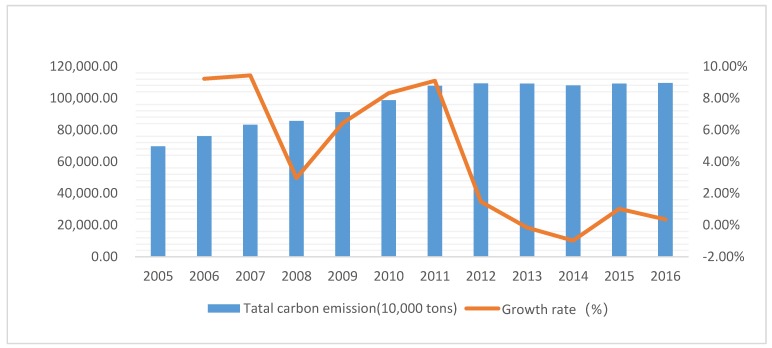
Growth trend of total carbon emission in the Yangtze River Economic Belt (YREB).

**Figure 2 ijerph-17-00545-f002:**
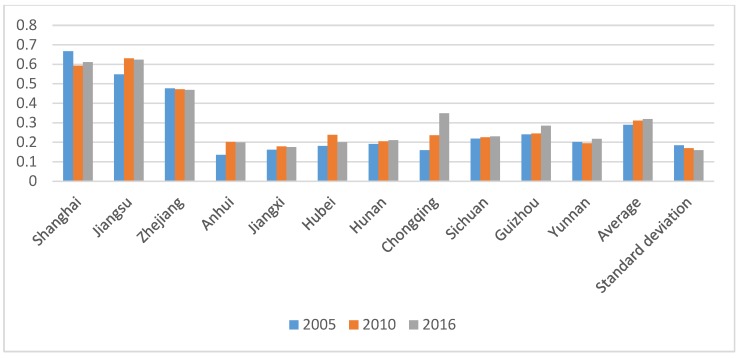
Comprehensive score of CERC.

**Figure 3 ijerph-17-00545-f003:**
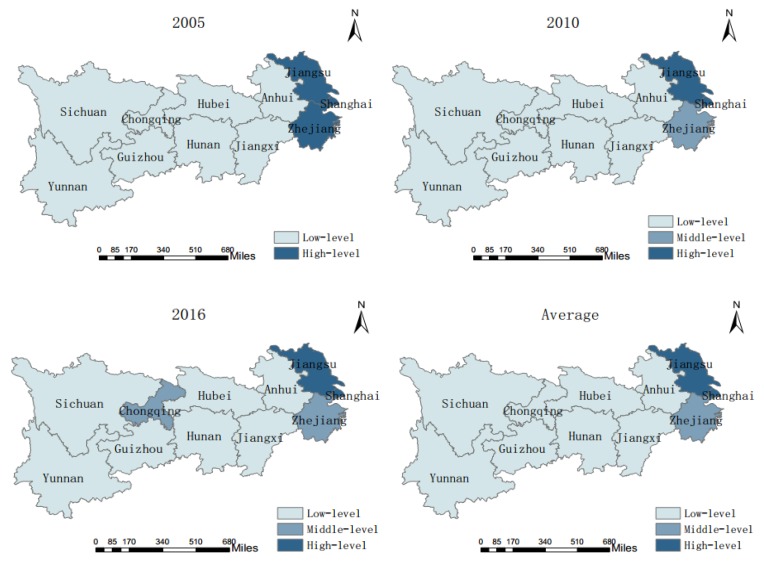
Spatial distribution of comprehensive scores of CERC of the YREB.

**Figure 4 ijerph-17-00545-f004:**
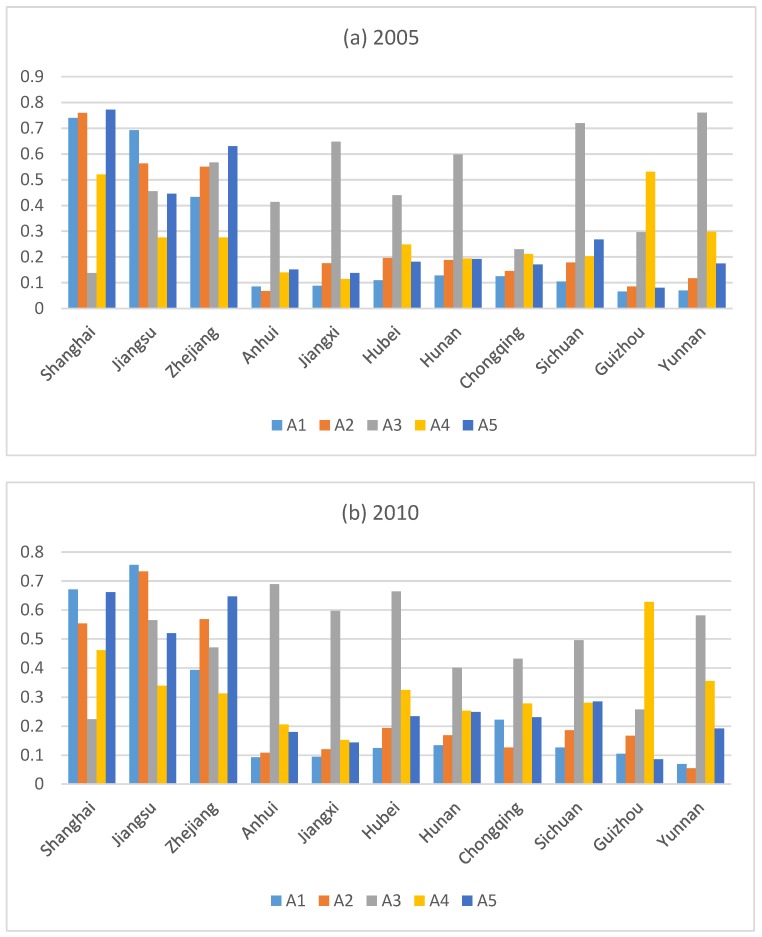
Carbon emission reduction subsystem score.

**Figure 5 ijerph-17-00545-f005:**
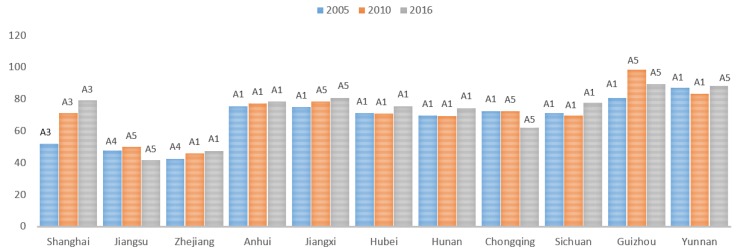
Substance factors and obstacles of CERC (%).

**Figure 6 ijerph-17-00545-f006:**
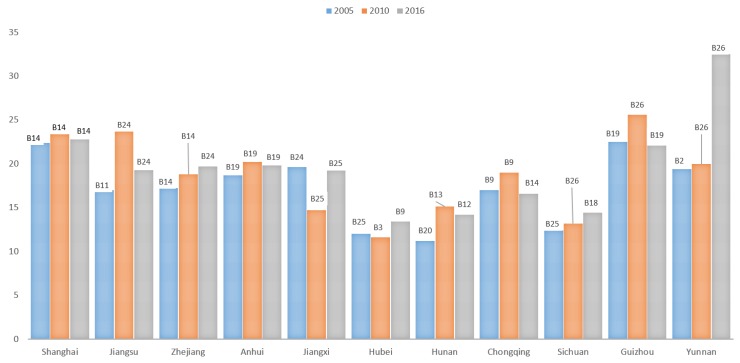
CERC index barrier factors and their obstacles (%).

**Figure 7 ijerph-17-00545-f007:**
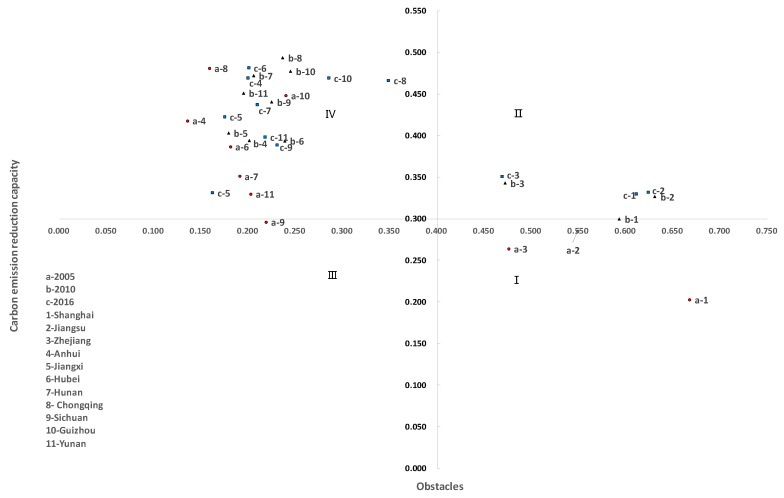
Four quadrants of carbon emission reduction potential in the YREB.

**Table 1 ijerph-17-00545-t001:** Supporting reference of evaluation index system.

Comment Content	Subsystem	Reference
emission reduction capacity	Industry and energy consumption structure Open to the outside world Technology and carbon sinks Energy consumption and carbon emissions Economic development	Yao et al. (2012) [13]
emission reduction potential	Carbon responsibility (Economic and social development, Energy and carbon efficiency) Carbon reduction capacity (Economic, technological, and carbon sink strength) Difficulties in carbon reduction	Wu et al. (2011) [14]
Low-carbon economy	Economic development and social progress Energy structure and usage efficiency Living consumption Development surroundings	Su et al. (2012) [8]
Low-carbon economy	Economy development social development Environmental assessment Science	Guo et al. (2017) [11]

**Table 2 ijerph-17-00545-t002:** Carbon emission reduction capacity (CERC) evaluation index system.

	Primary Indicator		Secondary Indicators	Unit	Attributes
A1	Economic development	B1	Gross domestic product divided by population	RMB	Positive
		B2	Gross domestic product growth rate	%	Positive
		B3	Urban per capita disposable income	RMB/year	Positive
		B4	Rural per capita net income	RMB/year	Positive
		B5	Fixed asset investment in the whole society	10^8^ RMB	Positive
		B6	Total import and export of goods by foreign-invested enterprises	10^4^ dollars	Positive
		B7	The tertiary industry accounts for the proportion of gross domestic product	%	Positive
A2	Science and technology	B8	Domestic patent grants	piece	Positive
		B9	Product quality	%	Positive
		B10	Technical market turnover	10^4^ RMB	Positive
A3	Carbon sink	B11	Forest cover rate	%	Positive
		B12	Per capita park green area	m^2^	Positive
		B13	Forest carbon uptake	10^4^ tons	Positive
		B14	Crop carbon uptake	10^4^ tons	Positive
A4	Energy consumption and carbon emission	B15	Energy intensity	tce/10,000 RMB	Positive
		B16	Carbon intensity	ton/10,000 RMB	Positive
		B17	Energy footprint	tce/person	Positive
		B18	Carbon Footprint	ton/person	Positive
		B19	Coal consumption as a share of energy consumption	%	Negative
B5	Social development	B20	Civil vehicle ownership	10^4^ cars	Positive
		B21	Urbanization rate	%	Positive
		B22	Urban residents’ consumption level	RMB/person/year	Positive
		B23	Rural residents’ consumption level	RMB/person/year	Positive
		B24	Teacher–student ratio in ordinary universities	Number of teachers = 1	Positive
		B25	Public transport vehicles per 10,000 people	Standard car	Positive
		B26	City gas penetration rate	%	Positive

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
