# Peer review of "A Comprehensive Evaluation of Carbon Emission Reduction Capability in the Yangtze River Economic Belt"

_ijerph, 2020, doi:10.3390/ijerph17020545_

Round 1

Reviewer 1 Report

This paper focus on 11 provinces and municipalities of the Yangtze River Economic Belt and use the TOPSIS method to assess their carbon emission reduction development status in the years of 2005, 2010 and 2016. The approach of this study is generally well-described. However, I don’t think the key conclusions/ messages are conveyed efficiently in the manuscript. Particularly, major conclusions are not shown in abstract. The presentation and description of some figures can be improved too. A few grammar issues are also needed to be fixed before publication.

Detailed comments:

Abstract: why not name the key obstacles in the abstract? the abstract should show the main conclusions of this study, which is currently not.

Line 30, ‘arrested’ ? do you mean ‘assessed’?

Line 33, and a few other places, does REF refers to missing citation?

Line 83, need to define ‘REC’.

Line 208, Shanghai and Zhejian are not in figure 1.

Fig.1, why use line plot when those cities are totally independent. I would recommend using bar plot for each year. What’s the y-axis stands for? Similarly for Fig. 3.

Line 211-213, In fig. 1, Chongqing shows the large increases instead of Anhui.

Line 223-224, what's the quantitative definition of high/low CERC level area?

Line 240-243, is this a speculation? If yes, consider changing the statement to reflect this. If not, please show the results of correlation analysis. 

Section 4.2.2 unclear how the subsystem is defined in this study. is it a sub-region of a city/province?

Fig. 6, unclear which city/province/year the black dot is representing. Consider using different color/symbols to make this figure more informative.

Author Response

This paper focus on 11 provinces and municipalities of the Yangtze River Economic Belt and use the TOPSIS method to assess their carbon emission reduction development status in the years of 2005, 2010 and 2016. The approach of this study is generally well-described. However, I don’t think the key conclusions/ messages are conveyed efficiently in the manuscript. Particularly, major conclusions are not shown in abstract. The presentation and description of some figures can be improved too. A few grammar issues are also needed to be fixed before publication.

Dear reviewers, thank you very much for your valuable advice. According to your proposal, we have made a substantial revision of the paper, so as to get your approval.

Point 1: Abstract: why not name the key obstacles in the abstract? the abstract should show the main conclusions of this study, which is currently not.

Response 1: Sorry, we have included the main conclusions of the study in the abstract on line 20-24.

Point 2: Line 30, ‘arrested’ ? do you mean ‘assessed’?

Response 2: “carbon emissions has been arrested”, this sentence translated into Chinese is “碳排放…被遏制了”. To make it easier to understand, we have replaced it with a more appropriate word.

Point 3: Line 33, and a few other places, does REF refers to missing citation?

Response 3: Sorry, Ref is the comment of this article, the comment is marked in the wrong place

and we have made a change.

Point 4: Line 83, need to define ‘REC’.

Response 4: Sorry, this is a spelling mistake, we have changed “REC” to “EC”.

Point 5: Line 208, Shanghai and Zhejiang are not in figure

Response 5: Sorry, we have made a change.

Point 6: Fig.1, why use line plot when those cities are totally independent. I would recommend using bar plot for each year. What’s the y-axis stands for? Similarly for Fig. 3.

Response 6: Sorry, we have made a change. Y-axis stands for obstacles to achieving the goal of carbon emission reduction.

Point 7: Line 211-213, In fig. 1, Chongqing shows the large increases instead of Anhui.

Response 7: Sorry, we have made a change.

Point 8: Line 223-224, what's the quantitative definition of high/low CERC level area?

Response 8: Page 230 outlines the classification criteria, and we use the standard deviation classification method to divide carbon emissions into three grades for each year

Point 9: Line 240-243, is this a speculation? If yes, consider changing the statement to reflect this. If not, please show the results of correlation analysis. 

Response 9: Sorry, we have changed the statement on line 272.

Point 10: Section 4.2.2 unclear how the subsystem is defined in this study. is it a sub-region of a city/province?

Response 10: There are 26 indicators in the indicator system, which are divided into 5 subsystems. The subsystem score is obtained by using the model to calculate the index data in the system. The indicator system can be applied to each province or municipality, so the subsystem is also applicable to each province or municipality.

Point 11: Fig. 6, unclear which city/province/year the black dot is representing. Consider using different color/symbols to make this figure more informative.

Response 11: Sorry, we have made a change.

Reviewer 2 Report

This study uses a entropy-based TOPSIS method to create a carbon emission reduction evaluation metric using several socio-economic and energy consumption factors. The goal of this study is to asses emission reduction capacities of YREB of China using this entropy-based method.

One major concern of this paper is:

- Line number 126 says, "design of the evaluation index system of low-carbon cities needs improvement" as one of the reasons for this study. Yet, no comparison with the state of the art method is provided to highlight improvement.  The impact of this study would be significantly more if such a comparison is provided.

Other particular comments are as follows:

1) In line 134, authors have talked about the entropy-based TOPSIS method before introducing the standard TOPSIS method (line 140). The standard method should be introduced first.
2) Line 192, what is the "factor analysis method"?
3) Why is the mean and standard deviation in the same point plot (Figure 1)? They indicate two different aspects of data and shouldn't be compared directly. Also, how is the standard deviation being computed in Figure 1? It is hard to compare the provinces on a line diagram, it would actually be quite informative to have them longitudinally ordered from east to west or reverse.
4) In line 207, what does it mean by "different range of increase and decrease"?
5) Line 239: nice comparison! Would it possible to perform a statistical test to check if the decrease is significant?

Author Response

This study uses a entropy-based TOPSIS method to create a carbon emission reduction evaluation metric using several socio-economic and energy consumption factors. The goal of this study is to asses emission reduction capacities of YREB of China using this entropy-based method.

Dear reviewers, thank you very much for your valuable advice. According to your proposal, we have made a substantial revision of the paper, so as to get your approval.

Point 1: Line number 126 says, "design of the evaluation index system of low-carbon cities needs improvement" as one of the reasons for this study. Yet, no comparison with the state of the art method is provided to highlight improvement. The impact of this study would be significantly more if such a comparison is provided.

Response 1: Thank you for your suggestion, we have added a comparison on line 205-216 and line 225.

Point 2: In line 134, authors have talked about the entropy-based TOPSIS method before introducing the standard TOPSIS method (line 140). The standard method should be introduced first.

Response 2: Thank you for your suggestion, we have added an introduction on line 147-149.

Point 3: Line 192, what is the "factor analysis method"?

Response 3: Sorry, this statement is not appropriate, we have adjusted the statement of indicator system on line 205-216 and line 225.

Point 4: Why is the mean and standard deviation in the same point plot (Figure 1)? They indicate two different aspects of data and shouldn't be compared directly. Also, how is the standard deviation being computed in Figure 1? It is hard to compare the provinces on a line diagram, it would actually be quite informative to have them longitudinally ordered from east to west or reverse.

Response 4: Thank you for your suggestion, we have modified the diagram, and the mean and standard deviation are just to describe the distribution of the data.
Point 5: In line 207, what does it mean by "different range of increase and decrease"?

Response 5: Sorry, we have made a change.
Point 6: Line 239: nice comparison! Would it possible to perform a statistical test to check if the decrease is significant?

Response 6: Sorry, since the sample data is small, I don't know how to test the significance of the decline.

Reviewer 3 Report

I think the significance of this thesis is relatively general, though the carbon emission of Yangtze economic belt has the strong research value. The analysis of this article is not thorough enough, and this article also does not explain the innovation and the unique features. The language of this article also needs to be improved.

1.Why the paper selects the area of Yangtze economic belt as the study area? I think it does not explain clearly in the introduction. The situation of carbon emissions in this region should be mentioned.

2. I think the paper does not consider the frontiers of international research, as most of references come from the Chinese journals. The literature analysis is incoherent, while many references are listed together simply.

3.TOPSIS should be one mature research method, so I think it is not necessary to explain the method too carefully, with 11 formulas. This section also do not consider the topic, so I wonder the value of this section.

4. The CERC evaluation index system contains many sectors, and I wonder whether so many indexs affect the CERC. How do you assume that the effect is positive or the effect is negative, without interrelated thesis?

5. The research period you selected is 2005-2016, but you mention the year of 2004. Also the period is 12 years, not 11 years. You only list the results of 2005, 2010, 2016, what about the other years?

6. The Yangtze economic belt contains 11 provinces (cities), but there 8 areas in your figures ,while some contain 11 areas. The analysis of these spatial differences only the descriptive statistical analysis of the results.

Author Response

I think the significance of this thesis is relatively general, though the carbon emission of Yangtze economic belt has the strong research value. The analysis of this article is not thorough enough, and this article also does not explain the innovation and the unique features. The language of this article also needs to be improved.

Dear reviewers, thank you very much for your valuable advice. According to your proposal, we have made a substantial revision of the paper, so as to get your approval.

Point 1: Why the paper selects the area of Yangtze economic belt as the study area? I think it does not explain clearly in the introduction. The situation of carbon emissions in this region should be mentioned.

Response 1: Sorry, we have added a description of carbon emissions from the YREB on Line 42-45 and provided the corresponding diagram on Line 54-55

Point 2: I think the paper does not consider the frontiers of international research, as most of references come from the Chinese journals. The literature analysis is incoherent, while many references are listed together simply.

Response 2: Six of the 28 articles in this article are from Chinese journals and 22 are from English journals. However, many articles published in English journals may be from Chinese authors. We have reorganized the documents.

Point 3: TOPSIS should be one mature research method, so I think it is not necessary to explain the method too carefully, with 11 formulas. This section also do not consider the topic, so I wonder the value of this section.

Response 3: Sorry, we have deleted the redundant formula.

Point 4: The CERC evaluation index system contains many sectors, and I wonder whether so many indexs affect the CERC. How do you assume that the effect is positive or the effect is negative, without interrelated thesis?

Response 4: Based on some relevant literature, we have decided to build an index system from five aspects: economic development, social development, scientific and technological development, environmental development and energy consumption. Among them, the positive index, that is, the increase of index value, promotes the improvement of evaluation result level, and the negative index, that is, the increase of index value, suppresses the improvement of evaluation result level. According to this principle, we refer to the relevant literature and track the clock to determine the positive and negative attributes of the index, and the relevant references have been stated in Table 1.

Point 5: The research period you selected is 2005-2016, but you mention the year of 2004. Also the period is 12 years, not 11 years. You only list the results of 2005, 2010, 2016, what about the other years?

Response 5: Because each year's data needs to be calculated separately, the 11-year calculation results are presented one by one, and the length will be too redundant, so the first and middle years between 11 years are selected for presentation. And we have changed “2005-2016” to “ in the years of 2005、2010、2016”.

Point 6: The Yangtze economic belt contains 11 provinces (cities), but there 8 areas in your figures, while some contain 11 areas. The analysis of these spatial differences only the descriptive statistical analysis of the results.

Response 6: Sorry, we have made a change.

Round 2

Reviewer 1 Report

    The revision has answered my comments and questions generally well. I would recommend minor edition on the English to make it more readable. 

Author Response

Response to Reviewer 1 Comments

The revision has answered my comments and questions generally well. I would recommend minor edition on the English to make it more readable. 

Dear reviewers, thank you very much for your valuable advice. According to your proposal, we have made a substantial revision of the paper, so as to get your approval.

Reviewer 3 Report

I think you have revised most of mistakes as I mentioned in the last review. But, I think there already have some places which should be explained. Fristly, you have mentioned the low-level in Fig.3, how did you distinguish the low, middle and high level. Secondly, this is January of 2020, and I think you can find the data of 2017 or 2018. If it is diffcult for you, Please explain why. Thirdly, In the analysis section, you should compare your results with the related paper, such as "A Study on The Driving Factors and Spatial Spillover of Carbon Emission Intensity in Yangtze River Economic Belt", which published in IJERPH just. However, there may be some related papers you should refer to.

Author Response

Response to Reviewer 3 Comments

I think you have revised most of mistakes as I mentioned in the last review. But, I think there already have some places which should be explained. Fristly, you have mentioned the low-level in Fig.3, how did you distinguish the low, middle and high level. Secondly, this is January of 2020, and I think you can find the data of 2017 or 2018. If it is diffcult for you, Please explain why. Thirdly, In the analysis section, you should compare your results with the related paper, such as "A Study on The Driving Factors and Spatial Spillover of Carbon Emission Intensity in Yangtze River Economic Belt", which published in IJERPH just. However, there may be some related papers you should refer to.

Dear reviewers, thank you very much for your valuable advice. According to your proposal, we have made a substantial revision of the paper, so as to get your approval.

Point 1: Fristly, you have mentioned the low-level in Fig.3, how did you distinguish the low, middle and high level.

Response 1: Because the evaluation score of carbon emission reduction is related to the selected index system and calculation method, there is no authoritative value for reference here. So the low, medium, and high levels here are relatively internal to the Yangtze River Economic Belt. For example, for the scores of the provinces and municipalities in 2005, we divide them into three effective intervals according to the standard deviation and average value of this group, and then we name the three intervals as low-level, medium-level and high-level zones according to the size of the interval values.

Point 2: Secondly, this is January of 2020, and I think you can find the data of 2017 or 2018. If it is diffcult for you, Please explain why.

Response 2: Because the index of crop carbon absorption is based on the index of crop yield, economic coefficient and carbon absorption rate, we have selected several kinds of crops, including grain, cotton, oil, hemp, vegetables and so on, shared by 11 provinces and municipalities in the Yangtze economic belt. The data for 2005,2010,2016 are taken from the official website of China Plantation Management Department, but the data can no longer be downloaded from the website in the second half of 2019, and there is no vegetable production in the main crop production of the China Statistical Yearbook, so sorry we didn't find the latest data.

Point 3: Thirdly, In the analysis section, you should compare your results with the related paper, such as "A Study on The Driving Factors and Spatial Spillover of Carbon Emission Intensity in Yangtze River Economic Belt", which published in IJERPH just. However, there may be some related papers you should refer to.

Response 3: Sorry, we have added some analysis on line 433-439 and line 446-455.
